# Sourcing Limestone Masonry for the Restoration of Heritage Buildings: Frumoasa Monastery Case Study

**DOI:** 10.3390/ma15207178

**Published:** 2022-10-14

**Authors:** Cătălin Onuțu, Dragoș Ungureanu, Dorina Nicolina Isopescu, Nicoleta Vornicu, Ionuț Alexandru Spiridon

**Affiliations:** 1Faculty of Civil Engineering and Building Services, “Gheorghe Asachi” Technical University of Iaşi, 43 Mangeron Blvd., 700050 Iaşi, Romania; 2Metropolitan Center of Research T.A.B.O.R., 9 Closca Str., 700066 Iaşi, Romania

**Keywords:** stone masonry, limestone, petrographic examination, XRF spectroscopy, heritage construction

## Abstract

Cultural and religious heritage assessments and restorations are considered to be a fundamental requirement of any modern society because these constructions represent one of the most meaningful and tangible connections to our past. With rare exceptions, heritage buildings were built with materials and systems that could bear gravitational loads but not bending and shearing resulting from seismic loading. Thus, in many cases, earthquake ground motions have led to severe degradation and even the collapse of various parts of these structural systems. In order to address these issues, repair and replacement techniques are applied as common parts of restoration work. In the peculiar case of stone masonry structures, a standalone macroscopic examination is not self-assured and, most often, can lead to an inadequate selection of a replacement material. Therefore, a knowledge of mesoscopic, petrographic, physical and mechanical properties is compulsory in the design, planning and execution of restoration work. From this perspective, the present research has taken, as a case of study, the Frumoasa monastic complex from Iași, Romania, introducing microscopic, XRF (X-ray fluorescence) spectroscopy and petrographically based approaches, comparing three limestone samples with a sample dislodged from the original wall. The physical properties (bulk and real densities, open porosity and capillary water absorption coefficient) and the mechanical properties (compressive and tensile strengths) were also experimentally determined. The samples were extracted from stone quarries located on the territories that were part of the same historical region as the Frumoasa monastic complex. Based on the outcomes of this study, suitable criteria for the stone replacement—consisting of identifying the main structure, quarry rock petrographical parameters and physical and mechanical characteristics—were determined and applied.

## 1. Introduction

Heritage constructions are witnesses to the cultural and religious history of society, helping to provide people with a sense of place and identity through a stable connection to the past. Thus, the rehabilitation and restoration of these buildings are nowadays considered to be compulsory actions to preserve their historical, religious, architectural, cultural and esthetic values. Among heritage constructions, masonry (made of bricks or stone) is almost always required for maintenance and restoration work [1,2,3]. Hence, a certain need exists to establish a methodology to ensure a trustworthy match between the existing in situ stones and the replacement ones. In Romania, this need is more acute when dealing with over 400 heritage structures and monuments, many of them being made whole or in part with stones ranging in color and composition [4].

The replacement of degraded stones in existing historic constructions is a deep-rooted practice in restoration work. Regardless of whether a strengthening system is to be applied or not, the masonry substrate should first be prepared and repaired to meet its initial constructive and appearance features [5,6,7]. In Europe, an ample debate has been held for decades on the priority and significance of the selection criteria for the materials used in restoration work [8,9,10]. On one side are the research teams who promote the traditional approach, devoted to saving the construction authenticity either by preserving the original materials or, in severe cases, entirely replicating the original structural system with initially used materials. On the other hand, there are research teams who are open to the use of new materials that are substantially different from the original ones, which may lead to considerable improvements to the overall structural behavior. This approach is valid only for materials that are compatible with the original ones from chemical and physical points of view. Nevertheless, both the conservative and the modern sides agree that any type of restoration work should be designed and applied in a suitable manner to avoid further worsening of the structural behavior [11]. Both sides also consider that the cultural and religious inheritance, the availability of the construction materials, the craftsmanship knowledge, the economic and the sustainable conditions should be evaluated in particular, case by case, to choose the appropriate materials and the type of restoration process.

The aim of this study was to determine and evaluate the mineralogic, petrographic, physical and mechanical properties of the stones used in the Frumoasa monastery complex from Iași, Romania, and their source areas to extract materials for the restoration work.

## 2. The Heritage and History of the Frumoasa Monastery Complex

The Frumoasa monastery complex is located in Iași, in the north-east part of Romania. The monastery was erected in 1587 and has been rebuilt twice; once between 1726 and 1733 and once between 1836 and 1839 [12]. According to the inscription from the church frontispiece, the current main building (The Church of The Holy Archangels Michael and Gabriel) was constructed between 1836 and 1839 by the abbot Ioasaf Voinescu: “This holy and divine church was built from the ground by the venerable archimandrite and knight Chir Ioasaf Anastasă, also known as Voinescu, with all his expenses, in honor of the Great Voivodes Michael and Gabriel and in honor of the Immaculate Assumption of the Mother of God, to the eternal memorial, praying for the last rulers of this monastery, to make a memorial on the days of these holidays and for the one who built this holy place. This building was started in 1836 and finished in 1839”.

The complex measures 14,906 m^2^ and consists of 5 distinct structures (The Church of The Holy Archangels Michael and Gabriel (built between 1836 and 1839), The Palace on The Walls (built between 1726 and 1733), The Ruins of The Palace for Women (built in the 18th century), The Bell Tower (built between 1819 and 1833) and The Enclosure Wall (built between 1726 and 1733)) (Figure 1). Between 1945 and 2002, all the structures except the main church were adapted and reused as a military hospital, barracks, prison and nursing school.

According to the structural assessment, due to the fact that the complex was used for a long time for a completely different purpose than what it was intended for as well as partial demolitions made between 1945 and 2002 and past earthquakes, the degradation state was severe. The latter consisted of ruined areas, partial demolitions, cracks, extended cracks, water infiltration and structural deficiencies. Moreover, due to the poor structural condition and water infiltration, the wall painting iconography was in jeopardy and the risk of non-remediable degradation was high. 

From the above-mentioned perspective, the degraded stone masonry needed to be repaired and reconstructed. Taking into consideration the religious, heritage and artistic value of the Frumoasa monastery complex, the restoration work of the church was performed with materials as close as possible to the 19th century original ones to preserve the construction authenticity in the best way possible. It is worth mentioning that the original materials from the degraded parts of the church walls were either missing or they were beyond any possibility of restoration.

## 3. Methodology

As mentioned before, Frumoasa monastery was built in 1587 and rebuilt twice, in the 18th and 19th centuries. In these periods, the city of Iași was part of The Principality of Moldavia. This country covered the region called Carpathian–Danube–Dniester, spanning from the Dniester River in the east to Transylvania in the west. Nowadays, medieval Moldavia is divided into three regions, part of three countries. These regions comprise eight counties of Romania, The Republic of Moldavia and the Budjak and Chernivtsi oblast areas of Ukraine. 

In the above-mentioned context, it was likely that the source materials used for the construction of Frumoasa monastery were extracted from stone quarries located on the territories that were part of the medieval Principality of Moldavia. Thus, for this study, samples were extracted from three stone quarries; two were located in Romania (Vama stone quarry and Poiana Deleni stone quarry) and one was located in The Republic of Moldavia (Egoreni) (Figure 2). The benchmark sample was dislodged from the original 19th century wall of The Church of The Holy Archangels Michael and Gabriel. The sample series and labeling are illustrated in Figure 3.

The microscopic, XRF (X-ray fluorescence) spectroscopy and petrographical studies were conducted in Iași, Romania, at the Metropolitan Center of Research T.A.B.O.R., a research facility devoted to studies regarding the rehabilitation and the reconstruction of heritage religious and cultural constructions. 

First, thin sections of 6 × 9 cm^2^ were prepared on all samples. The mineral compositions and textures were analyzed using an Olympus digital camera and shooting software corresponding with Olympus SZX 160 and U500XST2 microscopes. In addition, the XRF technique was implemented to determine the elemental composition of the samples. The measurements were realized using an isotope-free XRF spectrometer, type Innov X, combined with a mini PC.

In the second part of this experimental work, the quality of the samples was evaluated through the physical and mechanical characteristics of the rocks. The density and the open porosity were experimentally determined by testing five specimens from each sample series according to the specifications given in EN 1936 [15] and RILEM [16]. The same specimens were used to determine the water absorption coefficient by capillarity according to the specifications given in EN 1925 [17]. Uniaxial compression tests were performed on five specimens from each sample series using a WAW 600 computerized servo hydraulic universal testing machine. The tests were performed according to the recommendations of [18], imposing an axial displacement control rate of 10 μm/s. The tensile strengths of the rocks were indirectly determined by performing the Brazilian test according to the recommendations of [19]. Thus, 20 specimens (5 specimens for each sample series) were prepared and tested (using the same WAW 600 computerized servo hydraulic universal testing machine) at a loading rate of 200 N/s. 

## 4. Results

### 4.1. Petrography and XRF Spectroscopy

The main parameters that were identified and analyzed in the petrographic studies included the color, structure, texture, appearance and major minerals. These parameters were identified by a microscopic examination of thin sections of the samples. The petrographic characteristics of the samples are listed in Table 1. The images captured during the microscopic examinations are illustrated in Figure 4, Figure 5, Figure 6 and Figure 7.

The identification and analysis of the major and trace elements by XRF spectroscopy was possible due to the characteristic behavior of atoms when they interact with radiation [20,21]. When materials are excited by X-rays (high-energy and very short wavelength radiation), they can be ionized. 

The atoms become unstable when the energy of the X-ray is sufficient to dislodge a held close inner electron and an outer electron will replace the absent inner electron. When this phenomenon takes place, energy is released. The primary incident X-ray has a higher energy than the emitted radiation (named the fluorescent radiation). Due to the fact that the energy of the emitted photon in a particular element is characteristic of a transition between specific electron orbitals, fluorescent X-rays can be used to precisely detect the elements that exist in samples. The chemical composition of the samples analyzed in this study are listed in Table 2.

Based on optical microscopy and XRF spectroscopy, the S1 samples—dislodged from the original wall of Frumoasa monastery—contained calcium ions, positively charged potassium ions, mangan, iron in its +2 oxidation state, titanium and barium. Similar elements were found in the composition of the S2, S3 and S4 samples. The S2 sample contained calcium ions, positively charged potassium ions, titanium and pyrite. The difference between the S2 and S3 samples was that for the S3 sample series composition, mangan was also identified. The S4 samples contained calcium ions, positively charged potassium ions, pyrite, titanium and mangan. The XRF spectra of the analyzed samples are illustrated in Figure 8, Figure 9, Figure 10 and Figure 11.

The primary and secondary minerals found in the composition of the samples were highlighted by a microscopic examination (Figure 12). The dimensions of the primary components of the S1 sample, dislodged from the original wall of Frumoasa monastery, are listed in Table 3.

### 4.2. Physical Characteristics

As mentioned before, the density and the open porosity were experimentally determined by testing five specimens from each sample series according to the specifications given in EN 1936 [15] and RILEM [16]. Hydrostatic weighing was performed on dried specimens after the air voids were completely filled with water. Following that, the pore volume (that was accessible to water) was determined by applying the Archimedes principle, thus computing the porosity as well as the bulk and real densities. The presence of water is considered to be one of the main decaying factors of construction materials. Therefore, the capillary water absorption coefficient of the source limestone materials should be as low as possible. The capillary water absorption coefficient was determined as a function of the square root of time. The results are listed in Table 4.

### 4.3. Mechanical Characteristics

The results of the compression tests and the corresponding coefficients of variation are summarized in Table 5. As can be observed, there were a few specimens with more than twice the resistance determined for other stones belonging to the same sample series. This was proof of the quasi-homogeneity of the material, a fundamental characteristic for all the specimens, which was highlighted even if the stone samples of each series were cut from the same block of rock mass.

The results of the indirect tensile strengths, determined by Brazilian tests, and the corresponding coefficients of variation are summarized in Table 6. As can be observed, the coefficients of variation were high (> 15%), showing a large scattering of the results. This aspect was to be expected, considering the previously obtained values for the compressive strengths.

## 5. Discussion

All samples analyzed in this study were carbonate rocks belonging to the class of sedimentary rocks. The benchmark samples from the S1 series dislodged from the original wall of Frumoasa monastery were a variety of limestone and oolite and composed of grains of calcium carbonate (calcite) with dimensions ranging from 3 to 80 μm. The structure of these samples was weakened due to water infiltration. As a consequence, the minerals expanded in the presence of water and the permeability was high. Based on the outcomes of the microscopical and XRF analyses, the composition of the S3 samples (stone from Poiana Deleni stone quarry) was the closest to the composition of the benchmark S1 samples.

For all samples, the amount of clay minerals was less than 10% and the concentration of carbonates varied between 82% and 90%. As can be observed in Figure 3, the appearance of the samples was similar, except for the last specimen. The reddish-brown color of the S4 samples was due to the presence of iron and the venous texture was a consequence of multiple cracks in which the minerals could be found in very disparate amounts. 

By analyzing the experimental data for the physical and mechanical characteristics, we observed that there was an inverse relationship between the compressive strengths of the specimens and their capillary water absorption coefficient. The lowest average compressive strength (21.04 MPa) was obtained for the specimens belonging to the S1 sample series. For these specimens, the highest values for the average water absorption coefficient (4.5 kg/m^2^·h^0.5^) and the average open porosity (26%) were obtained. Similar unsatisfactory values were obtained for the specimens belonging to the S4 sample series. The low mechanical properties of these specimens were to be expected, considering the multiple cracks that were identified during the microscopic studies. On the other hand, the specimens belonging to the S3 sample series, in addition to the fact that they had the closest composition to one of the benchmark S1 samples, had the most favorable mechanical characteristics (average compressive strength: 35.49; average indirect tensile strength: 10 MPa). Moreover, the Poiana Deleni stone quarry (the provenance of the S3 sample series) is located close to the Frumoasa monastery complex (Figure 2).

The scattering of the experimental data related to the mechanical characteristics may have been the result of the non-homogeneous texture and structure of the specimens due to the mineral content as well as veins, discontinuities and fossil fragments. Similar or even higher values for the coefficients of variation have been reported in research works referring to limestone [22], calcareous stones [23], sandstones [24] and granite [25].

## 6. Conclusions

In Romania and Europe, most of the monumental constructions were built using different types of masonry systems. As a large share of this built heritage is in demand for repair, restoration and/or rehabilitation, there is a need to establish a methodology to ensure a trustworthy match between the existing in situ materials and the replacement ones. In the case of stone masonry, effective material matching requires both a petrological, microscopical and chemical analysis of the original material and a knowledge of the replacement options through the availability of stones from existing quarry sources in the settlement area. 

In the present study, a methodology was presented for selecting the appropriate replacement stone for the reconstruction of the Frumoasa monastery walls through microscopic, XRF (X-ray fluorescence) spectroscopy, petrographic, physical and mechanical analyses. Based on the main outcomes of this study, the following conclusions were drawn:The source materials used for the restoration of heritage constructions should be selected based on the importance class of the building, cultural and religious inheritance, availability of the construction materials, craftsmanship knowledge, economic and sustainable conditions.In the case of paramount heritage buildings, restoration work should be performed with original materials to preserve the construction authenticity as much as possible.The search for source materials should take into consideration the territorial divisions and organization from the time the heritage construction was built. Thus, samples should be extracted and analyzed from quarries located on the land that was once part of the settlement area.For this study, four sample series were analyzed: S1, a sample series dislodged from the original wall of Frumoasa monastery (Iași, Romania); S2, a sample series extracted from Egoreni stone quarry (Republic of Moldova); S3, a sample series extracted from Poiana Deleni stone quarry (Iași, Romania); and S4, a sample series extracted from Vama stone quarry (Suceava, Romania).All the analyzed samples were sedimentary rocks (limestone), mainly composed of calcium carbonate (CaCO_3_).The S1 samples were rocks with weakened aggregates due to internal and external degradation. The high concentration of potassium and iron was due to the clay minerals and the iron oxyhydroxides that formed the micrite.Based on the XRF (X-ray fluorescence) spectroscopy, the following carbonates were identified in the composition of the samples: CaCO_3_ (calcite); FeCO_3_ (siderite); and CaFe(CO_3_)_2_ (ankerite).As the esthetic factor is important when cladding internal and external walls and the apparent masonry as well as creating pedestrian walkways, the use of the S4 samples (Vama limestone) was not recommended. The iron oxides would permanently stain the surface in humid conditions and mineral expansion would occur, thus weakening the rock structure. Moreover, the hydrophobicity of this type of limestone would not make the existing vein texture disappear.The specimens belonging to the S3 sample series had a texture and appearance very similar to those of the benchmark S1 samples.The specimens belonging to the S3 sample series, in addition to the fact that they had the closest composition to the benchmark S1 samples, had the most favorable mechanical characteristics (average compressive strength: 35.49; average indirect tensile strength: 10 MPa).By analyzing all the outcomes of this study, it was concluded that the restoration work of the Frumoasa monastery walls should be performed with source materials extracted from the S3 quarry (Poiana Deleni stone quarry).

## Figures and Tables

**Figure 1 materials-15-07178-f001:**
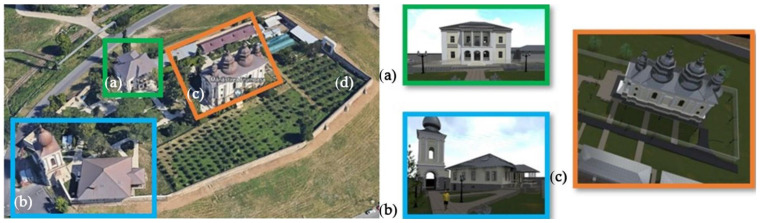
The Frumoasa monastery complex. (**a**) The Ruins of The Palace for Women; (**b**) The Bell Tower; (**c**) The Church of The Holy Archangels Michael and Gabriel and The Palace on The Walls; (**d**) The Enclosure Wall. Adapted from Google Earth [13].

**Figure 2 materials-15-07178-f002:**
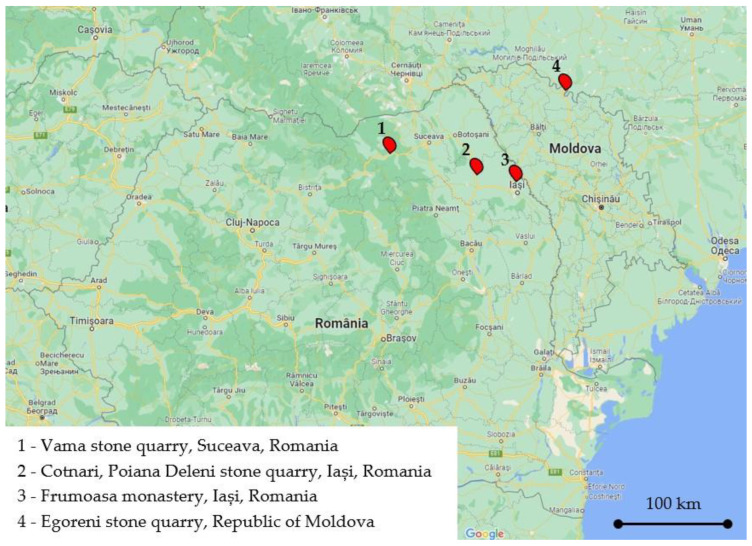
The location of Frumoasa monastery and the samples extracted as possible source materials for the reconstruction process. Adapted from Google Maps [14].

**Figure 3 materials-15-07178-f003:**
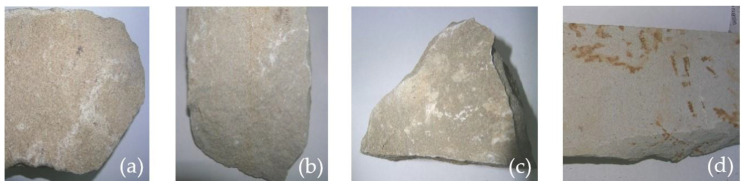
(**a**) S1: sample series dislodged from the original 19th century wall of The Church of The Holy Archangels Michael and Gabriel; (**b**) S2: sample series extracted from Egoreni stone quarry; (**c**) S3: sample series extracted from Poiana Deleni stone quarry; (**d**) S4: sample series extracted from Vama stone quarry.

**Figure 4 materials-15-07178-f004:**
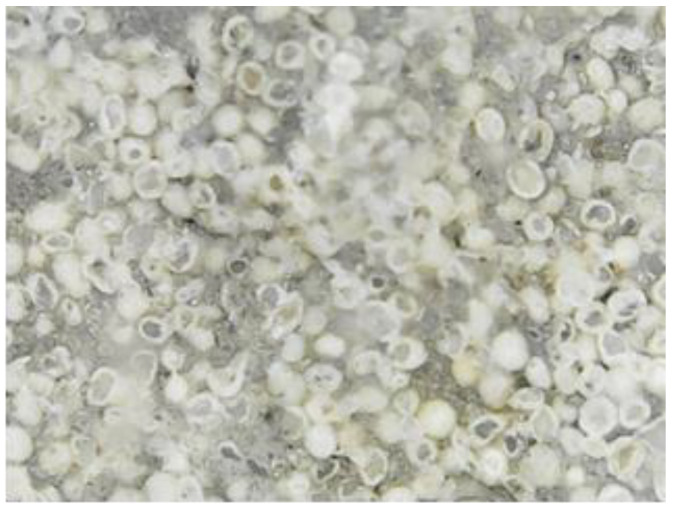
S1 sample series; magnification 500 X.

**Figure 5 materials-15-07178-f005:**
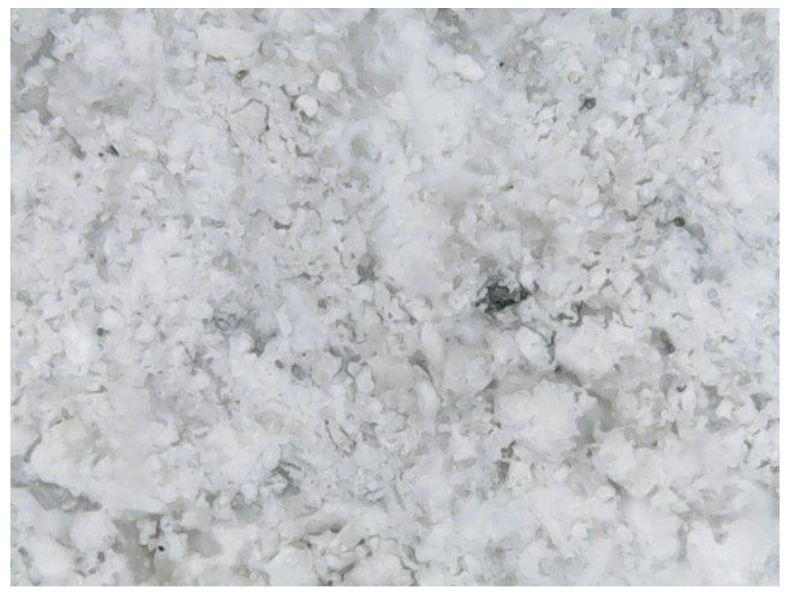
S2 sample series; magnification 500 X.

**Figure 6 materials-15-07178-f006:**
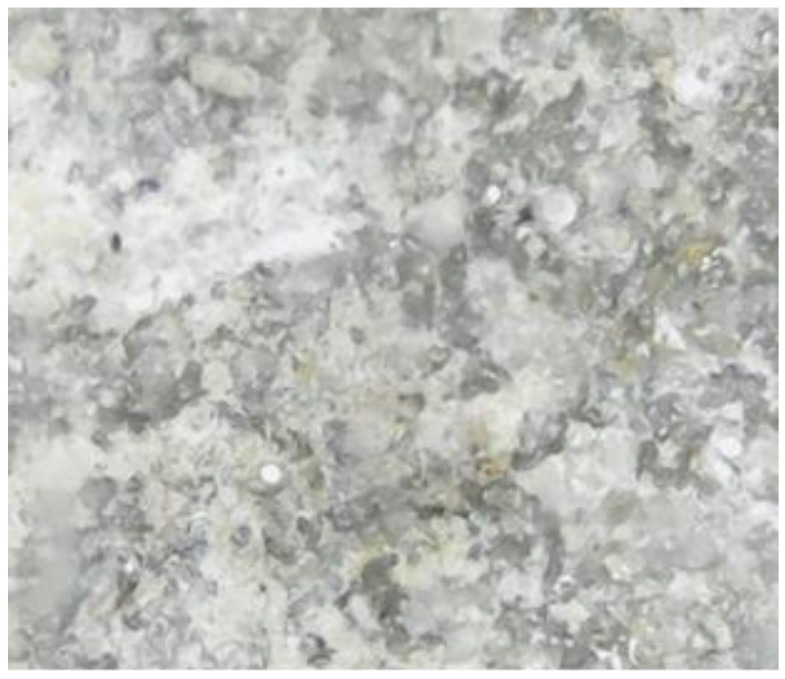
S3 sample series; magnification 500 X.

**Figure 7 materials-15-07178-f007:**
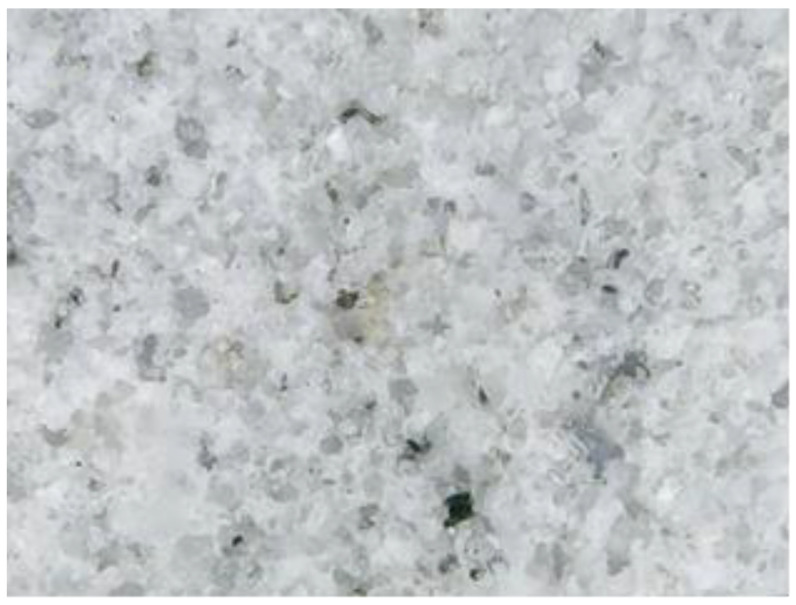
S4 sample series; magnification 500 X.

**Figure 8 materials-15-07178-f008:**
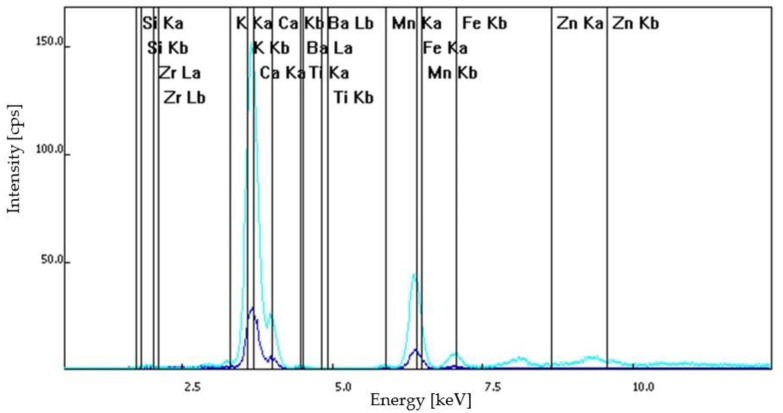
S1 sample series: XRF spectra.

**Figure 9 materials-15-07178-f009:**
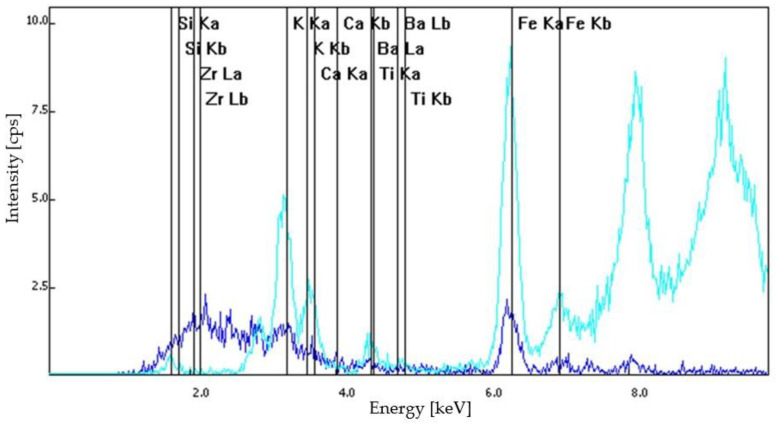
S2 sample series: XRF spectra.

**Figure 10 materials-15-07178-f010:**
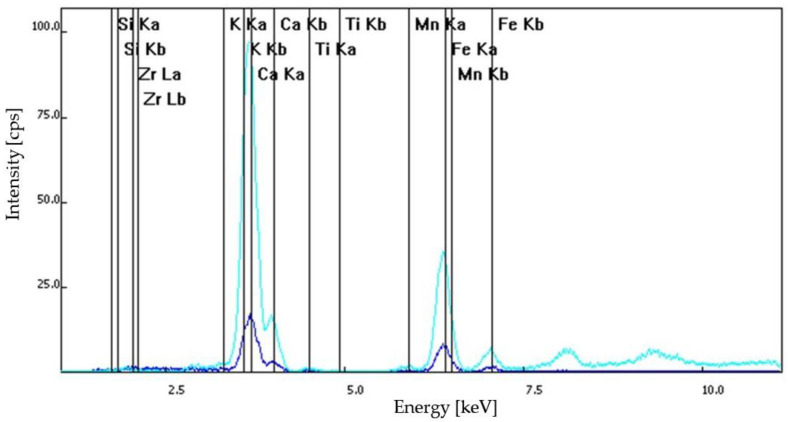
S3 sample series: XRF spectra.

**Figure 11 materials-15-07178-f011:**
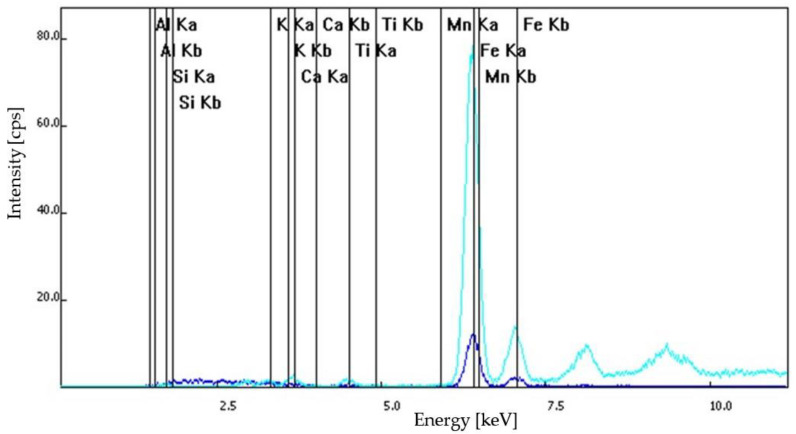
S4 sample series: XRF spectra.

**Figure 12 materials-15-07178-f012:**
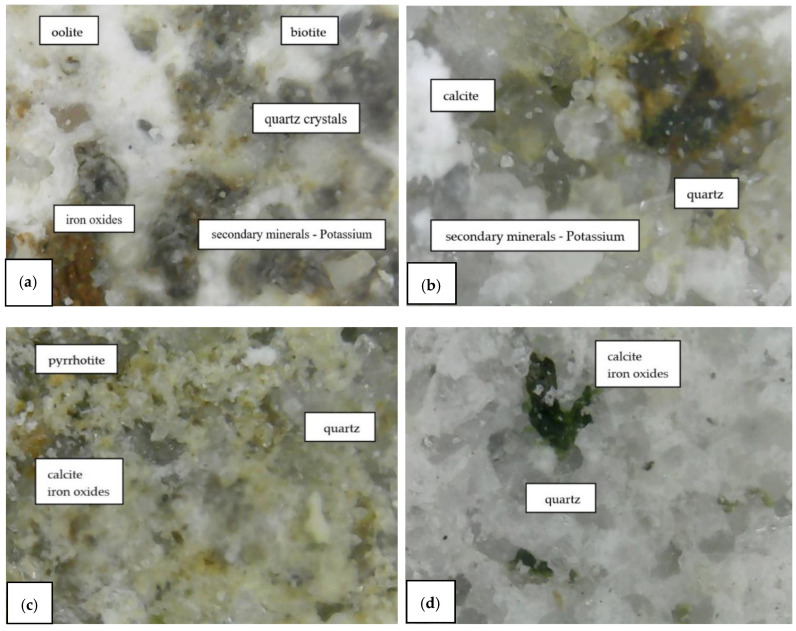
(**a**) S1 sample: magnification 500 X; (**b**) S2 sample: magnification 500 X; (**c**) S3 sample: magnification 500 X; (**d**) S4 sample: magnification 500 X.

**Table 1 materials-15-07178-t001:** The petrographic characteristics of the analyzed samples.

Characteristics	Results			
	S1	S2	S3	S4
**Structure**	Microcrystalline	Cryptocrystalline	Cryptocrystalline	Cryptocrystalline
**Texture**	Compact cytoplasmic autophagic vacuoles	Massive	Compact	Compact
**Appearance**	Heterogeneous	Homogeneous	Heterogeneous	Homogeneous
**Petrography**	Sedimentary rock	Sedimentary rock	Sedimentary rock	Sedimentary rock
**Major** **Minerals**	Calcite	Calcite	Calcite	Calcite

**Table 2 materials-15-07178-t002:** Chemical composition of the samples.

Sample Series	Main Elements (%)		Minority Elements (ppm)					
	CaO	K_2_O	Fe_2_O_3_	Ti_2_O	MnO	Ba	Zn	Zr
**S1**	25.2	2.36	14,414	1767	625	178	56	46
**S2**	0.51	2.95	1958	796	-	198	-	59
**S3**	29.5	1.54	9972	1756	603	-	-	36
**S4**	0.48	0.76	16,552	1615	-	-	31	118

**Table 3 materials-15-07178-t003:** The dimensions of the primary components of the S1 sample series.

Object Type	Measurement Type	Measurement No.	Value	Units	Description	Statistics		
						Magnitude	Length	Units
Line	Length	1	3	μm	Calcite grain	Number of measurements	4	μm
Line	Length	2	4	μm	Quartz	Average value	3.8	μm
Line	Length	2	12	μm	Silicon crystal	Average value	3.8	μm
Line	Length	3	80	μm	Calcite grain	Deviation	0.8	μm
Line	Length	2	2	μm	Halite	Average value	3.8	μm
Line	Length	3	8	μm	Quartz	Deviation	0.8	μm
Line	Length	4	32	μm	Silicon crystal	Number of measurements	4	μm

**Table 4 materials-15-07178-t004:** The physical characteristics of the samples.

Characteristics	Results			
	S1	S2	S3	S4
**Average open porosity (%)**	26	15	14	22
**Coefficient of variation (%)**	8.1	7.2	5.3	12.2
**Average bulk density (%)**	1203	2161	2312	1383
**Coefficient of variation (%)**	1.0	1.1	2.1	1.1
**Average real density**	1625	2542	2688	1773
**Coefficient of variation (%)**	0.3	0.1	0.2	0.1
**Average water absorption coefficient (kg/m^2^·h^0.5^)**	4.5	1.9	1.8	4.3
**Coefficient of variation (%)**	9.3	7.2	6.8	9.8

**Table 5 materials-15-07178-t005:** Compressive strength.

**S1 Sample Series**	**Results**				
	**S1.1**	**S1.2**	**S1.3**	**S1.4**	**S1.5**
**Compressive strength (MPa)**	22.13	18.11	27.12	13.34	24.51
**Average compressive strength (MPa)**	21.04				
**Coefficient of variation (%)**	25.82				
**S2 Sample Series**	**Results**				
	**S2.1**	**S2.2**	**S2.3**	**S2.4**	**S2.5**
**Compressive strength (MPa)**	36.21	35.65	21.14	41.03	37.81
**Average compressive strength (MPa)**	34.37				
**Coefficient of variation (%)**	22.36				
**S3 Sample Series**	**Results**				
	**S3.1**	**S3.2**	**S3.3**	**S3.4**	**S3.5**
**Compressive strength (MPa)**	37.82	38.21	34.68	39.12	27.61
**Average compressive strength (MPa)**	35.49				
**Coefficient of variation (%)**	13.27				
**S4 Sample Series**	**Results**				
	**S4.1**	**S4.2**	**S4.3**	**S4.4**	**S4.5**
**Compressive strength (MPa)**	25.12	25.11	14.65	28.20	18.31
**Average compressive strength (MPa)**	22.28				
**Coefficient of variation (%)**	25.10				

**Table 6 materials-15-07178-t006:** Indirect tensile strength.

**S1 Sample Series**	**Results**				
	**S1.1**	**S1.2**	**S1.3**	**S1.4**	**S1.5**
**Indirect tensile strength (MPa)**	2	4	2	6	3
**Average indirect tensile strength (MPa)**	3.4				
**Coefficient of variation (%)**	49.22				
**S2 Sample Series**	**Results**				
	**S2.1**	**S2.2**	**S2.3**	**S2.4**	**S2.5**
**Indirect tensile strength (MPa)**	5	8	10	9	8
**Average indirect tensile strength (MPa)**	8				
**Coefficient of variation (%)**	23.39				
**S3 Sample Series**	**Results**				
	**S3.1**	**S3.2**	**S3.3**	**S3.4**	**S3.5**
**Indirect tensile strength (MPa)**	10	11	9	8	12
**Average indirect tensile strength (MPa)**	10				
**Coefficient of variation (%)**	15.81				
**S4 Sample Series**	**Results**				
	**S4.1**	**S4.2**	**S4.3**	**S4.4**	**S4.5**
**Indirect tensile strength (MPa)**	4	4	2	6	2
**Average indirect tensile strength (MPa)**	3.6				
**Coefficient of variation (%)**	46.48				

## Data Availability

The data underlying this article will be shared on reasonable request from the corresponding author.

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
