# Peer review of "Sourcing Limestone Masonry for the Restoration of Heritage Buildings: Frumoasa Monastery Case Study"

_materials, 2022, doi:10.3390/ma15207178_

Round 1

Reviewer 1 Report

line 47 - Preserver authenticity means first of all preserving the material authenticity and therefore applying all those restoration thecniques for the conservation of the material and not its replacement. This is always an extreme choice when nothing else can be done.

lines 49-50 New materials can also cause serious damage to the structure if they are incompatible from a chemical or physical point of view.

line 86 What are the original materials in a historical monument? They are all original materials both those of the first constructions and post earthquake repairs. preserving autenticity means preserving what has come down to us in its material consistency.

line 88 - It is not clear whether the monastery is made up of parts built in different periods or has been completely rebuilt and therefore today is a 19th century monastery

It is not clear where the sample was taken in the monastery, whether in a part of the 19th century or in an earlier part. What are you comparing? This is important for any subsequent historical and economic reasoning.

Author Response

We appreciate your precious time in reviewing our paper and providing valuable comments. It was your valuable and insightful comments that led to possible improvements in the current version. The authors have carefully considered the comments and tried our best to address every one of them. We hope the manuscript after careful revisions meet the publication requirements of Materials Journal. The authors welcome further constructive comments if any. Below we provide a point-by-point response. All modifications in the manuscript have been highlighted in yellow.

line 47 - Preserver authenticity means first of all preserving the material authenticity and therefore applying all those restoration thecniques for the conservation of the material and not its replacement. This is always an extreme choice when nothing else can be done

  • Thank you for pointing this out. The text was revised. Please see rows 52-54

Lines 49-50 New materials can also cause serious damage to the structure if they are incompatible from a chemical or physical point of view.

  • Thank you for pointing this out. The text was revised. Please see lines 56-58.

Line 86 What are the original materials in a historical monument? They are all original materials both those of the first constructions and post earthquake repairs. preserving autenticity means preserving what has come down to us in its material consistency.

  • Thank you for pointing this out. The text was revised. Firstly, the components of the Frumoasa monastery complex were described. Please see lines 72-85. After that, the material state was explained. Please see lines 100-104.

Line 88 - It is not clear whether the monastery is made up of parts built in different periods or has been completely rebuilt and therefore today is a 19th century monastery

  • The main building of the monastery (The Church of The Holy Archangels Michael and Gabriel) was completely reconstructed between 1836-1839. Please see the revised text (lines 72-80).

It is not clear where the sample was taken in the monastery, whether in a part of the 19th century or in an earlier part. What are you comparing? This is important for any subsequent historical and economic reasoning.

  • We agree with the reviewer’s assessment. Accordingly, throughout the manuscript, we have revised the text and explain clearly the location of the samples (lines 118-119; 137-138).

Author Response

The authors appreciate the time and effort that the reviewer dedicated to providing feedback on our manuscript and are grateful for the insightful comments that provided valuable improvements to our paper. We hope the revised manuscript will suit the Materials Journal and we are available for further revisions if needed. Please see below a point-by-point response to the reviewer’ comments and concerns.

English should be revised. There are several word misuses throughout the manuscript.

  • Thanks for your kind reminders. We went through the entire manuscript to eliminate grammatical mistakes. Please see the yellow highlighted lines.

There is no conclusion regarding the practical aspects and applicability of this detail. Contrary

to the mentioned aim of this study, there is no methodology for selecting the appropriate

replacement stone for the reconstruction of the Frumoasa monastery walls. Although the work is

unique and has direct use in restoration of the studied heritage building, the application is too

limited and innovation is not highlighted. Also, the study is too narrowed to XRF. No other

studies from XRD, SEM was made in the material level. No physical (e.g. mass density, water

absorption, etc.) and mechanical properties (e.g. compressive strength, tensile strength. Modulus

of elasticity, etc.) was determined. No further studies on the building e.g. existing durabilityrelated damages to the material, the meteorological features of the site, etc. are missing.

We agree with the reviewer’s assessment. Thus, new sections were added:

  • the components of the monastery complex were better described (lines 72-80; 82-85),
  • the material state was explained (lines 100-104),
  • the samples’ locations were indicated (lines 118-119; 137-138),
  • new experimental work related to the physical properties of the samples (bulk and real densities, open porosity and capillary water absorption coefficient) were added and described (lines 141-153; 214-226),
  • new experimental work related to the mechanical properties of the samples (compressive and tensile strengths) were added and described (lines 146-154; 227-241),
  • new sections, part of the selecting methodology, were added (lines 247-249; 256-274)
  • the conclusion section was enriched (lines 319-328).
  • new references were added (lines 365-366; 373-379; 385-394)

Reviewer 3 Report

The paper focuses on a case of study the Frumoasa monastic complex from Iași, Romania, introducing microscopic, XRF (X-ray fluorescence) spectroscopy and petrographically based approaches, comparing four limestone samples with the sample extracted from the original wall. According to results, suitable criterion for stone replacement, consisting in identifying the main structure and quarry rock petrographically parameters, was determined and applied.

CONCLUSION:

In the article, microscopic, XRF (X-ray fluorescence) spectroscopy and petrographic properties of limestone samples taken from 4 different regions were evaluated. However, the referee thinks that the evaluation made is limited. The article does not include any conclusions about the mechanical and durability strengths of the material. Consequently, in my opinion the paper cannot be published in your journal.

Author Response

The authors appreciate the time and effort that the reviewer has dedicated to providing valuable feedback on the manuscript. We believe that the revised version of our paper addresses all concerns by the reviewer in detail. We hope the revised manuscript will suit the Materials Journal and we are available for further revisions if needed. Please see below our detailed response to comments.

  1. In the article, microscopic, XRF (X-ray fluorescence) spectroscopy and petrographic properties of limestone samples taken from 4 different regions were evaluated. However, the referee thinks that the evaluation made is limited. The article does not include any conclusions about the mechanical and durability strengths of the material. Consequently, in my opinion the paper cannot be published in your journal.

We agree with the reviewer’s assessment. Thus, new sections were added:

  • the components of the monastery complex were better described (lines 72-80; 82-85),
  • the material state was explained (lines 100-104),
  • the samples’ locations were indicated (lines 118-119; 137-138),
  • new experimental work related to the physical properties of the samples (bulk and real densities, open porosity and capillary water absorption coefficient) were added and described (lines 141-153; 214-226),
  • new experimental work related to the mechanical properties of the samples (compressive and tensile strengths) were added and described (lines 146-154; 227-241),
  • new sections, part of the selecting methodology, were added (lines 247-249; 256-274)
  • the conclusion section was enriched (lines 319-328).
  • new references were added (lines 365-366; 373-379; 385-394)

Reviewer 4 Report

In this paper, a stone sample is taken from a wall inside a masonry heritage complex, and then its properties in terms of minerals, XRF, and petrography are compared to three other samples taken from three different areas nearby. The idea is to develop a methodology to be able to select a repairing/replacement material as consistent as possible with the original one in a heritage structure.

The manuscript is written in a clear and concise way, and it is worthy to be published on the basis of the novelty and applicability offered in the paper. The English is easy to follow, and therefore, the paper is recommended for publication. The minor points raised by the reviewer are as follows:

1)      Abstract: In Line 25, it is mentioned that four samples are compared with a sample extracted from the original structure. First point is that if there are three samples or four samples compared to the original one? Based on Figure 3, it seems that three samples are compared with the original one. Secondly, it is advised to add one sentence in Abstract, explaining what are those other samples? In other words, it can be added that those additional samples are taken from some nearby quarry sources.

2)      Line 89, and Line 196: “was build” should be replaced by “was built”.

3)      Lines 99-101: Three samples taken from different areas are explained but the original sample taken from the original wall in the case study under consideration was left with no explanation in the text. So, it is advised to add a sentence explaining the sample S1 for clarification when referring to Figure 3.

4)      The main conclusion as stated in Lines 232-234 needs to be addressed earlier somewhere in the main body of the manuscript (in the sections before Conclusions). Perhaps, it is a good idea to add some explanations in this regard under Section 6 (i.e., Discussion).

5)      In “Introduction”, Lines 50-52, the following paper is recommended to be added to the list of references:

Nasrollahzadeh, K., & Zare, M. (2020). Experimental investigation on axially loaded adobe masonry columns confined by polymeric straps. Construction and Building Materials262, 119895.

Author Response

The authors appreciate the time and effort that the reviewer has dedicated to providing valuable feedback on the manuscript. We believe that the revised version of our paper addresses all concerns by the reviewer in detail.

 In this paper, a stone sample is taken from a wall inside a masonry heritage complex, and then its properties in terms of minerals, XRF, and petrography are compared to three other samples taken from three different areas nearby. The idea is to develop a methodology to be able to select a repairing/replacement material as consistent as possible with the original one in a heritage structure.

The manuscript is written in a clear and concise way, and it is worthy to be published on the basis of the novelty and applicability offered in the paper. The English is easy to follow, and therefore, the paper is recommended for publication. The minor points raised by the reviewer are as follows:

1)      Abstract: In Line 25, it is mentioned that four samples are compared with a sample extracted from the original structure. First point is that if there are three samples or four samples compared to the original one? Based on Figure 3, it seems that three samples are compared with the original one. Secondly, it is advised to add one sentence in Abstract, explaining what are those other samples? In other words, it can be added that those additional samples are taken from some nearby quarry sources.

  • Thank you for pointing this out. The text was revised. Please see lines 24-31; 114-115, 118-119; 137-140

Line 89, and Line 196: “was build” should be replaced by “was built”.

  • Thanks for your kind reminders. We went through the entire manuscript to eliminate grammatical mistakes. Please see the yellow highlighted lines.

Lines 99-101: Three samples taken from different areas are explained but the original sample taken from the original wall in the case study under consideration was left with no explanation in the text. So, it is advised to add a sentence explaining the sample S1 for clarification when referring to Figure 3

  • Thank you for pointing this out. The text was revised. Please see lines 118-119, 137-138.

The main conclusion as stated in Lines 232-234 needs to be addressed earlier somewhere in the main body of the manuscript (in the sections before Conclusions). Perhaps, it is a good idea to add some explanations in this regard under Section 6 (i.e., Discussion).

  • We agree with the reviewer’s assessment. Thus, new explanations were added. Please see lines 247-249; 256-274; 319-327.

In “Introduction”, Lines 50-52, the following paper is recommended to be added to the list of references:

Nasrollahzadeh, K., & Zare, M. (2020). Experimental investigation on axially loaded adobe masonry columns confined by polymeric straps. Construction and Building Materials, 262, 119895.

  • We agree with the reviewer’s assessment. Since we find valuable information for this study, we cited the paper. Please see lines 59-60; 365-366.

Round 2

Reviewer 2 Report

The authors improved the quality of the manuscript.